# Inhibition of Very Long Chain Fatty Acids Synthesis Mediates PI3P Homeostasis at Endosomal Compartments

**DOI:** 10.3390/ijms22168450

**Published:** 2021-08-06

**Authors:** Yoko Ito, Nicolas Esnay, Louise Fougère, Matthieu Pierre Platre, Fabrice Cordelières, Yvon Jaillais, Yohann Boutté

**Affiliations:** 1Laboratoire de Biogenèse Membranaire, Université de Bordeaux, CNRS, 33140 Villenave d’Ornon, France; yoko.ito@u-bordeaux.fr (Y.I.); Nicolas.Esnay@unt.edu (N.E.); louise.fougere@u-bordeaux.fr (L.F.); 2BioDiscovery Institute, Department of Biological Sciences, University of North Texas, Denton, TX 76203, USA; 3Laboratoire Reproduction et Développement des Plantes, Université de Lyon, ENS de Lyon, UCB Lyon1, CNRS, INRAE, 69342 Lyon, France; mplatre@salk.edu (M.P.P.); yvon.jaillais@ens-lyon.fr (Y.J.); 4Plant Molecular and Cellular Biology Laboratory and Integrative Biology Laboratory, Salk Institute for Biological Studies, La Jolla, CA 92037, USA; 5Bordeaux Imaging Center, Université de Bordeaux, CNRS, 33000 Bordeaux, France; fabrice.cordelieres@u-bordeaux.fr

**Keywords:** very-long chain fatty acids, phosphatidylinositol-3-phosphate, trans-Golgi network, multivesicular bodies, endosomes, vacuole

## Abstract

A main characteristic of sphingolipids is the presence of a very long chain fatty acid (VLCFA) whose function in cellular processes is not yet fully understood. VLCFAs of sphingolipids are involved in the intracellular traffic to the vacuole and the maturation of early endosomes into late endosomes is one of the major pathways for vacuolar traffic. Additionally, the anionic phospholipid phosphatidylinositol-3-phosphate (PtdIns (3)P or PI3P) is involved in protein sorting and recruitment of small GTPase effectors at late endosomes/multivesicular bodies (MVBs) during vacuolar trafficking. In contrast to animal cells, PI3P mainly localizes to late endosomes in plant cells and to a minor extent to a discrete sub-domain of the plant’s early endosome (EE)/trans-Golgi network (TGN) where the endosomal maturation occurs. However, the mechanisms that control the relative levels of PI3P between TGN and MVBs are unknown. Using metazachlor, an inhibitor of VLCFA synthesis, we found that VLCFAs are involved in the TGN/MVB distribution of PI3P. This effect is independent from either synthesis of PI3P by PI3-kinase or degradation of PI(3,5)P_2_ into PI3P by the SUPPRESSOR OF ACTIN1 (SAC1) phosphatase. Using high-resolution live cell imaging microscopy, we detected transient associations between TGNs and MVBs but VLCFAs are not involved in those interactions. Nonetheless, our results suggest that PI3P might be transferable from TGN to MVBs and that VLCFAs act in this process.

## 1. Introduction

In eukaryotic cells, cellular functions are extensively supported by membrane trafficking which sorts and targets proteins and lipids into distinct destinations. The Golgi apparatus and post-Golgi compartments such as the trans-Golgi network (TGN) are major sorting stations in eukaryotic cells. In plants, the TGN is a highly complex compartment which serves as a platform for both endosomal and secretory pathways where sorting mechanisms are differentiated to define trafficking routes to either the plasma membrane, the late endosomes (LE)/multivesicular bodies (MVBs), vacuolar compartments or other organelles [1,2]. A particularity of plant cells is that the TGN is the early endosome (EE) and can separate from the Golgi apparatus and move independently from it (Golgi-independent TGN; GI-TGN) [3,4]. Moreover, a general consensus is now emerging on the existence of distinct sub-domains at the TGN. Morphologically, at least two populations of vesicles are observed, one is apparently coatless, structured in round vesicles connected by membrane tubules and called Secretory Vesicles (SVs), although this name does not necessarily reflects their function as they can host endocytic materials; another one is coated with clathrin and called Clathrin-Coated Vesicles (CCVs) [3,5,6,7]. SVs host several protein sorting machineries such as the transmembrane protein ECHIDNA and its interactors, the Rab GTPase-interacting YIP4A/B proteins, which are involved in differential secretion of the auxin carrier AUX1 and PIN3, but not PIN2, and cell wall components [6,8,9]. Additionally, SVs host lipid-mediated protein sorting machineries given that SVs are enriched in sphingolipids containing Very-Long-Chain Fatty Acids (VLCFAs) and that the acyl-chain length of sphingolipids is involved in TGN-sorting of PIN2, but not PIN1 or AUX1, at SVs [10]. TGN-associated CCVs have been described to come in part from endocytosis; due to that, they are only partially uncoated after scission from the plasma membrane, and could act in the endocytic recycling between the plasma membrane and the TGN [11,12,13]. Other data have suggested that clathrin could act in the trafficking from the TGN to vacuoles but this idea is still a matter of debate [14,15,16]. Indeed, it has recently been observed that the adaptor protein 4 (AP4) labels a sub-domain of the TGN that co-localizes with the SNARE (soluble N-ethylmaleimide-sensitive factor attachment protein receptors) protein VAMP727 that is involved in vacuolar trafficking via MVBs while AP1 labels a sub-domain of the TGN that co-localizes with the SNARE protein VAMP721 that functions in secretory trafficking to the plasma membrane [17,18,19,20]. Interestingly, clathrin is not present at the AP4-mediated vacuolar trafficking zone but is detected at a distal side of AP1-positive SVs that detach from the rest of the Golgi apparatus [17]. Hence, CCVs could be a differentiation from GI-TGN. Interestingly, no vesicular budding was observed from the AP4-mediated vacuolar trafficking zone, suggesting an alternative path to transport cargos from the TGN to the vacuole [16,17]. Previously, it was suggested that the pre-vacuolar compartment MVBs mature from the TGN; this has been supported by electron microscopy observations of localization of Endosomal Sorting Complex Required for Transport (ESCRT I to III) and pharmacological treatment that inhibits MVB formation [21]. Furthermore, it was proposed that MVB maturation occurs at a specific sub-domain of the TGN which recruits the Rab5-like ARA7 concomitantly with a transient accumulation of the anionic phospholipid phosphatidylinositol-3-phosphate (PI3P) [22,23]. ARA7 mostly accumulates at MVBs, but a secondary pool has been described to localize at the sub-domain of the TGN, labeled the V-ATPase VHA-a1 [22,24]. Similarly, in *Arabidopsis*, fluorescent biosensors have revealed that PI3P is mainly localized to MVBs and vacuoles with a sparse pool are also present at VHA-a1-positive sub-domain of the TGN [25,26]. PI3P is required to recruit the ARA7 effector EREX and the ESCRT component FYVE1/FREE1 to late endosomes [27,28]. However, the mechanisms controlling the relative distribution of PI3P at endosomal compartments are unknown and the significance of PI3P at the TGN is still largely unclear. In this study, we show that VLCFAs are involved in the homeostasis of PI3P between TGN and MVBs. High resolution live cell imaging airyscan microscopy revealed that the TGN and PI3P-positive MVBs can transiently associate. However, our results indicate that VLCFAs do not act in TGN/MVBs association dynamics. While the quantification of PI3P levels at the compartments specifically during the association time was technically too challenging to address in this study, we used a pharmacological approach to additionally suggest that the pool of PI3P at the TGN could be transferable to MVBs. Altogether, our results show that VLCFAs, which are predominant in the pool of sphingolipids, are involved in the homeostasis of PI3P at endosomal compartments.

## 2. Results

### 2.1. Very Long Chain Fatty Acids Impact Sub-Cellular PI3P Homeostasis and Distribution

In *Arabidopsis* root, PI3P mainly resides at LE/MVBs [25,26]. To check whether VLCFAs could impact the PI3P localization pattern we used metazachlor, a chemical allowing a fine-tunable reduction of C24- and C26-Very-Long Chain Fatty Acids (VLCFAs), which are predominant in the pool of sphingolipids [10]. To visualize PI3P we used two genetically-encoded biosensors specific to PI3P; the first one is the 1xPX domain of the P40 protein fused to mCITRINE (1xPX^P40^-mCIT) [26]; the second one is the 2xFYVE domain of the HRS protein fused to mCITRINE (mCIT-2xFYVE^HRS^) [26]. Using exactly the same acquisition settings in control and metazachlor-treated samples, we found that metazachlor strongly reduces the intensity of the fluorescence of both PI3P biosensors at intracellular structures (Figure 1a,b,e,f). However, when we enhanced the signal on post-acquisition images, we could see the intracellular dots with a weak fluorescence in metazachlor-treated samples, and we did not quantify any change in the number of intracellular compartments (Figure 1a,c,e,g). Moreover, as a control we quantified the global intensity of fluorescence of biosensors in a given region of interest (the intensity then corresponds to intracellular structures and cytoplasm) and could not see any changes of intensity upon metazachlor treatment (Figure 1d,h). These results suggest that the decrease of PI3P biosensors at intracellular compartments is correlated to a disassociation of the probe from the membranes and is not due to a decrease in the number of intracellular structures, a general decrease of biosensor quantity or a decrease of the expression of biosensors upon metazachlor treatment. Thus, our results indicate that VLCFAs play a role in PI3P homeostasis.

Next, we checked whether the co-localization degree between PI3P-labeled compartments and either the TGN or MVBs was altered upon metazachlor. To this end, given that PI3P was described to mainly localize at MVBs and sparsely at the EE/TGN compartment, we performed co-localization analyses between PI3P-labeled compartments and either the MVB marker RAB-G3f fused to mCHERRY or the TGN marker VHA-a1 fused to mRFP [29,30]. While it was shown that 1xPX^P40^-mCIT and mCIT-2xFYVE^HRS^ reside both mostly at MVBs and weakly at the TGN, it was also described that 1xPX^P40^-mCIT additionally labels the vacuoles, resulting in greater dilution of this biosensor over the intracellular membranes [26]. Thus, we used mCIT-2xFYVE^HRS^ for co-localization analysis with the TGN but were unsuccessful in getting the mCIT-2xFYVE^HRS^ biosensor crossed line with the MVB marker RAB-G3f fused to mCHERRY during the course of this study. Instead, we used the 1xPX^P40^-mCIT biosensor for co-localization analysis with MVBs. As a consequence, the co-localization degree of PI3P biosensors with the TGN or MVBs is not directly comparable. Nonetheless, our results revealed a significant decrease of PI3P-positive compartments co-localization with MVBs and a significant increase of co-localization with VHA-a1-positive sub-domain of the TGN when the root cells were treated with metazachlor (Figure 2a–d).

It should be stated here that the co-localization method we used is an object-based approach that accounts for the morphology of the compartments, independently from the intensities at these compartments. Consequently, our results show that more TGN compartments were PI3P-positive upon metazachlor treatment, but this does not necessarily imply an increase of PIP3 in each TGN. Indeed, we were not able to address the intensity values specifically at TGN given that, for co-localization analyses, we adjusted the acquisition settings for each sample so that we could get similar intensity at compartments between control and metazachlor-treated seedlings to avoid biases in the morphology of the compartments.

Additionally, during the course of co-localization analyses we noticed that the fluorescent signal of mCHERRY-RAB-G3f was weaker at MVBs and more cytosolic upon metazachlor treatment (Figure 2e). We quantified the intensity of fluorescence of mCHERRY-RAB-G3f at intracellular dots and found that the intensity significantly decreased upon metazachlor treatment (Figure 2f). After the maturation of MVBs from the TGN, it is thought that the conversion of the Rab5-like ARA7 into the Rab7-like RAB-G3f occurs at MVBs to assist MVBs-vacuole fusion [22,31]. Thus, our results indicate that VLCFAs impact the localization of one RAB-GTPase that has been shown to act downstream of the maturation of MVBs from the PI3P-enriched sub-domain of the TGN. Altogether, our results indicate that VLCFAs act on PI3P homeostasis and the relative distribution of PI3P between the TGN and MVBs. Thus, we next wondered whether the TGN and MVBs, although being two distinct endomembrane compartments, could be found sometimes in close association and whether VLCFAs would be involved in regulating this association.

### 2.2. The TGN Transiently Associates with PI3P-Positive Compartments but VLCFAs Are Not Involved in This Process

To evaluate whether the VHA-a1 subdomain of the TGN could associates with PI3P-positive structures we performed high resolution Airyscan microscopy on *Arabidopsis* roots expressing both VHA-a1-mRFP and mCIT-2xFYVE^HRS^. By time-lapse observation with Airyscan, we found that some mCIT-2xFYVE^HRS^-positive dots were closely located to VHA-a1-labeled TGN and move together for a while (Figure 3a). To get a general idea on whether we could detect association between the TGN and PI3P-positive structures we first used the distance analysis (DiAna) plugin of ImageJ on single-shot images. We calculated that the X-Y limit of resolution in our condition (mRFP and mCIT) was about 140 nm (λ = emission wavelength of mCIT, NA = numerical aperture of the objective lens, X = resolution improvement ratio by Airyscan [32]; 0.61 ∗ λ/NA/X = 0.61 ∗ 529/1.3/1.7). We also determined that the average distance between centroids of the TGN and centroids of PI3P-positive structures, for which we could get enough resolution to clearly state that the two structures are separated, is 620 nm. Thus, we considered that an association event between the TGN and PI3P-positive dots is located within a 140–620 nm resolution window. Using these settings, we could detect association events between the TGN and PI3P-positive compartments (Figure 3b). However, metazachlor treatment did not alter the number of associations between the TGN and PI3P-positive compartments (Figure 3b). Nevertheless, as we performed these quantifications on a single time point it is possible that the associations we detected were just coincidental. We reasoned that if real associations occur between the TGN and PI3P-positive dots then they will last for a certain amount of time while coincidental associations would not. Hence, we set up a new plugin on ImageJ called “Detect, Track and Colocalize” (DTC) to track individual objects in two colors on 2D time lapse movies and check for co-localization or proximity/associations between both channels over a tracked period of time (https://github.com/fabricecordelieres/IJ-Plugin_DTC, accessed on 27 July 2021). Our results indicate that the TGN and PI3P-positive compartments can remain associated for an average time of 6 s (Figure 3c). In comparison, PI3P-positive compartments that are not associated with the TGN could be followed unassociated for an average time of 19 s (Figure 3c). However, our quantifications showed that metazachlor does not alter the dynamics of association between the TGN and PI3P-positive compartments as the number of associations or the time of association per track is not different on metazachlor-treated roots as compared to untreated controls (Figure 3d).

Given that PI3P mainly localizes to MVBs and sparsely at TGN, we can reasonably assume that PI3P-positive compartments which do not contain the TGN marker VHA-a1 are MVBs. Together, our results indicate that the TGN and PI3P-positive MVBs can transiently associates for 6 s in average but VLCFAs are not involved in this process. However, we do not exclude that VLCFAs could be involved in PI3P exchange or homeostasis at the TGN/PI3P-positive MVBs interface independently of an association process. To test this hypothesis, it would be important to measure the fluorescence intensity of PI3P biosensors specifically during the time of association between the TGN and PI3P-positive MVBs. Unfortunately, this approach was too challenging to be addressed during the course of this study. Instead, we employed a pharmacological approach in order to deplete the pool of PI3P at MVBs and further investigate the effect of metazachlor on the intracellular distribution of PI3P biosensors.

### 2.3. Metazachlor Prevents the Loss of PI3P upon Inhibition of the Phosphatidylinositol 3-Kinase

The fungal metabolite wortmannin is an inhibitor of the phosphatidylinositol 3-kinase (PI3K) complex that produces PI3P at MVBs [33,34]. In *Arabidopsis* roots, wortmannin causes homotypic fusion of MVBs resulting in typical donut-shaped MVBs agglomeration [31,34,35]. Wortmannin inhibits PI3K at 1 µM while it inhibits both PI3K and PI4K at 33 µM [33,36,37]. While the use of 1 µM wortmannin would be more specific to PI3P, it has been shown previously that this concentration does not have any effect on the morphology, mobility or distribution of the PI3P-positive MVBs [34]. Thus, we chose to work with 33 µM wortmannin and first checked the PI4P localization pattern upon wortmannin. In the control seedlings, the PI4P biosensor mCIT-3xPH^FAPP1^, was mainly localized at the PM and weakly in intracellular dots after 5 min of wortmannin treatment (5 min, Figure 4a). While the intensity of the mCIT-3xPH^FAPP1^ at the PM decreased already after 15 min of wortmannin treatment and became even more weaker at 90 min of treatment, the fluorescence intensity at intracellular dots was not modified even after 90 min treatment (Figure 4b,c). We previously found that metazachlor treatment causes the accumulation of PI4P at TGN [38], this accumulation was not modified by additional wortmannin treatment (Figure 4c). These results show that wortmannin impacts PI4P quantity at the PM but not at intracellular compartments.

In wild-type control seedlings, the PI3P mCIT-2xFYVE^HRS^ biosensor signal intensity at intracellular compartments was not significantly different at the beginning of the wortmannin treatment (5 min, 15 min and 45 min), although the signal intensity tends to decrease, consistently to what has been published before [39] (Figure 4d,e). These results suggest that a compensation mechanism exists to recover PI3P upon inhibition of the PI3K. Indeed, after 90 min of wortmannin treatment, the mCIT-2xFYVE^HRS^ signal intensity was much stronger at donut-like intracellular compartments (Figure 4d,e). Interestingly, in metazachlor grown seedlings, mCIT-2xFYVE^HRS^ signal intensity gets higher already after 45 min of wortmannin treatment, much faster than in control seedlings (90 min, Figure 4d,e). Moreover, we also noticed that the donut-shaped MVBs agglomeration appear earlier (Figure 4d). These results indicate that metazachlor treatment could compensate the loss of PI3P upon inhibition of the PI3K. An explanation for this observation could be that the TGN serves as a small sink of PI3P. Previously, it was shown that upon wortmannin, homotypic fusion of MVBs as well as heterotypic fusion of the TGN and MVBs occur [34]. Given that the PI3K complex is located at MVBs and that more TGNs contain PI3P biosensor upon metazachlor, it is possible that the increased number of TGN containing PI3P accounts for the compensation of metazachlor-treated seedlings to PI3P-loss upon wortmannin treatment.

Another explanation could be that, upon metazachlor and wortmannin treatment the production of PI3P from PI is enhanced, which is highly unlikely given that wortmannin inhibits PI3K, or that the degradation of PI(3,5)P_2_ into PI3P gets enhanced.

### 2.4. VLCFA-Mediated PI3P Homeostasis Is Not Dependent upon PI(3,5)P_2_ Degradation by SAC1

Faster recovery of the PI3P biosensor mCIT-2xFYVE^HRS^ upon wortmannin in metazachlor-treated seedlings could come from the consumption of the pool of PI(3,5)P_2_ through the PI(3,5)P_2_ phosphatase activity of the SAC1-5 family [40,41,42]. We reasoned that if the consumption of PI(3,5)P_2_ occurs faster in metazachlor grown seedlings, the recovery of PI3P during wortmannin treatment should be delayed in a PI(3,5)P_2_ phosphatase mutant. We chose a mutant for the *SAC1* gene as the SAC1 protein has a PI(3,5)P_2_ phosphatase activity and the corresponding *sac1* mutant displays plant growth defects at the single mutant stage while SAC2-5 need to be combined in multiple mutants to display phenotypic defect [41,42]. We introduced the mCIT-2xFYVE^HRS^ PI3P biosensor into the *sac1* homozygote background and performed wortmannin treatment as described above. In contrast to wild-type, in the *sac1* mutant mCIT-2xFYVE^HRS^ intensity significantly decreased from 5 min to either 15 or 45 min of wortmannin treatment (Figure 5a,b). These results suggest that SAC1 is involved in the compensation mechanism triggered upon PI3K inhibition, most probably through the generation of PI3P from PI(3,5)P_2_. However, when we treated the *sac1* mutant with metazachlor, we observed that mCIT-2xFYVE^HRS^ signal intensity gets higher already after 45 min of wortmannin treatment, much faster than in control seedlings (90 min, Figure 5a,b). These results are not different from wild-type experiments and suggest that the SAC1 phosphatase, although involved in a PI3P compensation mechanism, is not involved in mediating the effect of VLCFAs on PI3P level. SAC1 was found to localize at the TGN where it could regulate PI3P homeostasis at the TGN/PI3P-positive MVBs interface [41]. We constructed a stable pUBQ10::Venus-SAC1 *Arabidopsis* expressing line and found that Venus-SAC1 localizes at the tonoplast consistently to the localization of its close homologs SAC2-5 (Figure 5c) [42]. Thus, consumption of PI(3,5)P_2_ probably occurs at the vacuole/MVB interface but is not involved in VLCFA-dependent PI3P homeostasis at the TGN/PI3P-positive MVB interface.

### 2.5. PI3P of TGN Might Be Transferable to MVBs

Given that we observed an increase of PI3P biosensors at the TGN upon metazachlor, one possibility is that the pool of PI3P at the TGN serves as a small sink that recovers PI3P upon wortmaninn. In support of that hypothesis, in metazachlor-treated seedlings, we observed a decrease of the co-localization level between the PI3P biosensor mCIT-1xFYVE^HRS^ and the TGN marker VHA-a1-mRFP after 15 min, as compared to 5 min, of wortmannin treatment (Figure 6a,b). These results suggest that the pool of PI3P at the TGN may be transferrable to MVBs and that VLCFAs may act in this process.

## 3. Discussion

In eukaryotic cells, the maturation of the LE occurs from the EE through a process called Rab conversion that switches the small GTPase Rab5 into Rab7 when promoted by the PI3P-binding complex Ccz1-SAND/Mon1 and the vacuolar protein sorting HOPS complex [23,43,44]. In plants, LE/MVB matures from the EE/TGN and more particularly from a sub-domain of the TGN labeled by VHA-a1 and locally enriched in PI3P [21,22]. While the SAND/Mon1 is also involved in Rab conversion in plants, it does not function in the maturation of MVBs but rather acts in the fusion of MVBs with vacuoles [22,23]. Whether PI3P at the TGN is required for TGN to MVBs maturation is not known in plants. However, PI3P is essential for protein sorting at MVBs as, for example, the FYVE1/FREE1—a component of the ESCRT complex that sorts membrane ubiquitinated cargos into intraluminal vesicles of MVBs—localizes at MVBs through interaction with PI3P [28]. Moreover, PI3P is required to recruit EREX, an effector of the Rab5-like ARA7, to endosomes [27].

Our study identifies that VLCFAs are involved in TGN/MVBs PI3P balance as shortening the acyl-chain length of fatty acids results in a higher amount of PI3P-containing TGNs and a lower amount of PI3P-containing MVBs. Interestingly, we did not observe any differences in the number of MVBs upon acyl-chain length shortening, indicating that VLCFA-mediated PI3P homeostasis is not involved in MVB maturation from the TGN. These results are consistent with the fact that PI3P-binding SAND1/Mon1 is not involved in MVB genesis from the TGN [22]. In the future, it would be interesting to test whether VLCFAs are involved in PI3P-mediated protein sorting by mediating the localization of ESCRT components and/or altering the sorting of ubiquinated cargos at MVBs. Indeed, Rab5-to-Rab7 conversion occurs during MVB maturation and the subsequent fusion of MVBs with the vacuole partly through the Guanosine Exchange Factor (GEF) activity of SAND1/Mon1 which activates RAB-G3f, a member of the Rab7 family [22,23,31]. Interestingly, our results indicate that VLCFAs act on the regulation of the quantity of RAB-G3f at MVBs. Thus, it would be worth testing in the future whether VLCFAs are involved in SAND1/Mon1 localization as well.

Our results show that the VLCFA-regulated pool of PI3P is unlikely to be correlated to synthesis from PI or degradation from PI(3,5)P_2_. While previous work suggested that the PI(3,5)P_2_ phosphatase SAC1 localizes at TGN, we could not confirm this result and rather confirmed a localization to the tonoplast, as described for SAC2-5 [41,42]. Thus, it appears that the SAC1-5 PI(3,5)P_2_ phosphatases act more in the MVBs/vacuole interface than the TGN/MVBs interface. Our results further suggest that SAC1 is involved in a PI3P compensation mechanism when the PI3K is inhibited. Thus, PI3P at MVBs could potentially come from the consumption of PI(3,5)P_2_ by SAC1 at the vacuole. However, our results indicate that SAC1 is not involved, at least not alone, in VLCFA-mediated PI3P homeostasis.

As VLCFAs are involved in the homeostasis of PI3P at endosomal compartments, we postulated that they could regulate PI3P homeostasis during potential TGN/PI3P-positive MVB membrane associations. While we could detect transient TGN/PI3P-positive MVB association events, lasting for an average of 6 s, we did not observe any involvement of VLCFAs in this dynamics of association. Nevertheless, we do not exclude that VLCFAs could regulate PI3P levels specifically during TGN/PI3P-positive MVB associations while having no function in the establishment of these associations per se. In animal cells, lipid exchange through membrane contact sites have been evidenced especially in the case of PI4P/sterol exchange [45,46]. It is unlikely that this sterol-exchange mechanism is controlled by VLCFAs in plant cells as reducing the acyl-chain length by metazachlor does not alter sterol content in microsomal or TGN-purified fraction [10,38]. We could not test this hypothesis further during the course of this study due to the technical challenge of measuring fluorescence intensity variations specifically during TGN/PI3P-positive MVB associations. However, using wortmannin, an inhibitor of the PI3K activity that depletes PI3P at MVBs and causes MVB aggregation in donut forms [31,34,35], we could detect that when we shorten the acyl-chain length of fatty acids the quantity of PI3P-positive TGNs decreased and the apparition of donut-like structures of agglomerated MVBs appeared earlier. These results suggest that PI3P could potentially be transferred from TGN to MVBs. Nevertheless, this is only an assumption that would remain to be tested further by other approaches in the future. Moreover, we do not rule out that the effects we see could be due to several independent VLCFA-mediated processes, one could be the consumption of PI3P, for which we had the fluorescence intensities of biosensors as a readout, and another one could be localization balance between the TGN and PI3P-positive MVBs, for which we had the co-localization analyses as a readout.

VLCFAs are mostly predominant in the pool of sphingolipids and to a lesser extend in phosphatidylserine (PS) [10,47,48,49]. While VLCFAs of sphingolipids are known to be involved in vacuolar trafficking, the involvement of PS in vacuolar pathways is unknown [50,51,52]. Our results most probably imply a function of sphingolipids in the regulation of PI3P homeostasis at endosomal compartments, but it will be important in the future to decipher further the role of VLCFA-sphingolipids and/or PS in this process.

## 4. Materials and Methods

### 4.1. Plant Material and Growth Conditions

The following *Arabidopsis thaliana* transgenic fluorescent protein marker lines were used: PI3P biosensors pUB10::1xPH^p40^-mCITRINE (P3Y) [26], pUB10::mCITRINE-2xFYVE^HRS^ (P18Y) [26]. PI4P biosensor pUBQ10::mCITRINE-3xPH^FAPP1^ [39]. MVB marker pUB10::mCHERRY-RAB-G3f (W5R) [29]. TGN marker pVHA-a1::VHA-a1-mRFP [30]. The double fluorescent lines were established by crossing of the lines above. Sterilized seeds were kept at 4 °C in water for 2–3 days, sown on half Murashige and Skoog (MS) agar medium plates (0.8% plant agar, 1% sucrose, and 2.5 mM morpholinoethanesulfonic acid pH5.8 with KOH), and grown in 16 h light/8 h darkness at 22 °C for 5 days.

### 4.2. Cloning and Plant Transformation

The *pUB10::Venus-SAC1* was cloned by Multisite Gateway^®^ Three-Fragment Vector Construction Kit (ThermoFisher, Waltham, Massachusetts, MA, USA. First, the genomic sequence of *SAC1* (AT1G22620) was recover by PCR using the PrimeStar Max DNA Polymerase (Takara Bio, (Kusatsu, Shiga, Japan)) and the following primers: attB2r-SAC1 5′- GGGG ACA GCT TTC TTG TAC AAA GTG GGG atg gcg aaa tcg gaa aac tc -3′and attB3-SAC1 5′- GGGG AC AAC TTT GTA TAA TAA AGT TGG tta aat gac ctt cgg gac c -3′ including a gene specific sequence fused to the attB2r or attB3 sequences for proper BP recombination. The BP reaction was set according to the manufacturer’s protocol to insert *SAC1* in the pDONR™ P2r-P3 and then transformed in DH5α and the cells were propagated on LB plates with 50 µg/mL of Kanamycin. After the sequence was controlled by sequencing, the LR reaction was set using the following entry clones: P4-P1r-pUb10 containing the Ubiquitin 10 promotor, the P1-P2-Venus containing the Venus for an N-terminus fusion to *SAC1* and P2r-P3-SAC1, and the destination vector pK7m34GW [53]. After an overnight recombination, the vector was transformed in DH5α and then the cells were propagated on LB plates containing a 1:1 mix of Spectinomycin and Streptomycin at 100 µg/mL. The final vector was transformed in C58C1 *Agrobacterium tumefaciens* by heat shock to later be used for plant transformation.

The plants were transformed by floral dip [54] during which the flowers were dipped twice for 30 s in the transformation solution (5% sucrose, 0.05% silwet L-77) containing the agrobacteria bearing the *pUB10::Venus-SAC1* plasmid. After the floral dip, the plants were wrapped in plastic bags to maintain a high humidity for 24 h; afterwards, the bags were opened to decrease the humidity slowly for 72 h. After senescence, the seeds were harvested and sown on a half MS plate containing Kanamycin at 35 µg/mL and the growing plants were then moved into a pot to produce the T2 seeds.

### 4.3. Inhibitor Treatments

Mz treatment was performed on seedlings grown for 5 days on half MS plates containing 50 or 100 nM Mz (Cayman Chemical, Ann Arbor, Michigan, MI, USA). Mz was added to the medium from a 100 mM stock in dimethylsulfoxide by using an intermediate diluted stock at 100 µM (extemporarily prepared). Wm treatment was performed on seedlings grown on drug-free or Mz-containing half MS plates for 5 days and transferred into a liquid half MS medium containing 33 µM of Wm (Merck, Darmstadt, Germany) with or without Mz, and the treated plants were served for observations after the time indicated in the figures.

### 4.4. Confocal Microscopy and Image Analyses

Confocal laser scanning microscopy was performed using a Zeiss LSM 880. Seedlings were mounted with 1/2 MS medium (with or without drugs). Double-sided tape was used as the spacer to separate the slide glass and coverslip. All acquisitions were done with an oil-immersion ×40 objective, 1.3 numerical aperture (APO 40×/1.3 Oil DIC UV-IR).

Quantification of the fluorescence intensity was performed by ImageJ. For the PM, the outline of the cell was drawn outside of the PM by hand, and the signal intensity was quantified in the region that is within 1.5 µm inside the outline, which was subsequently normalized by the area. For the intracellular dots, mask images were created by applying the threshold, and those masks were used to extract the dots from the original images. The total signal intensity was normalized by the number of the dots. The threshold was kept constant for all the samples that are shown in the same graph. In order to avoid including a non-dotty background, only the structures with circularity over 0.1 were quantified (circularity is defined as 4*πA/P*^2^ with *A* = area and *P* = perimeter, and it takes the values 0.0–1.0 with 1.0 representing the perfect circle).

Co-localization analyses in Figure 2; Figure 5 were performed using the geometrical (centroid) object-based method [55]. Subcellular compartments were extracted by applying a threshold, and the distance between the centroids of two objects was calculated using 3D objects counter plugin of imageJ. For the analysis of Figure 5, images were additionally processed by applying subtract background (rolling ball radius = 10 pixels, approximately 1.3 µm) in order to get better segmentation of the dots. When the distance between two labelled structures was below the optical resolution limit, the co-localization was considered as true. The resolution limit was calculated based on the shorter emission maximum wavelength of the fluorophores.

The analysis of association between intracellular dots in fixed images (Figure 3b) was performed by ImageJ plugin DiAna [56]. Before analyzing, a Gaussian blur filter (0.1 µm radius) was applied followed by subtracting background (rolling ball radius = 10 pixels, approximately 1.3 µm). The center–center distance was quantified by DiAna, and the number of PI3P labeled dots that have the closest TGN within the range from 1 pixel (approximately 125 nm) to 5 pixels (approximately 625 nm, the sum of average radius of TGN and PI3P dots) was counted.

For the analysis of association between intracellular dots in time-lapse movies (Figure 3c,d), a newly developed ImageJ plugin “Detect, Track and Colocalize” (DTC) (https://github.com/fabricecordelieres/IJ-Plugin_DTC, accessed on 27 July 2021) was used. Before the analysis, Gaussian blur filter and subtracting background was applied similarly to the fixed image analysis. The PI3P dots that appear in the movie for 3 frames (4 s) in minimum were tracked with 10 pixels (approximately 1.3 µm) maximum displacement per 2 s. Within the tracks, the number of frames that have the closest TGN within the range from 1 pixel (approximately 125 nm) to 5 pixels (approximately 625 nm) and the number of continuous sequences of those frames were quantified.

### 4.5. Statistical Analyses

For the comparison of two groups, two-sided Wilcoxson’s rank-sum test was used. Kruskal–Wallis test followed by Dwass–Steel–Critchlow–Flinger multiple comparison test was used for the comparison of 3 and more groups. All the statistics were performed with R (version 3.6.0 and RStudio (version 1.2.1335. Variances between each group of data are represented in boxplot, bee swarm or dot plot. Each element of the boxplot indicates the following value: Center line, median; box limits, the first and third quartiles; whiskers, 1.5× interquartile range; points above or below the whiskers, outliers.

## Figures and Tables

**Figure 1 ijms-22-08450-f001:**
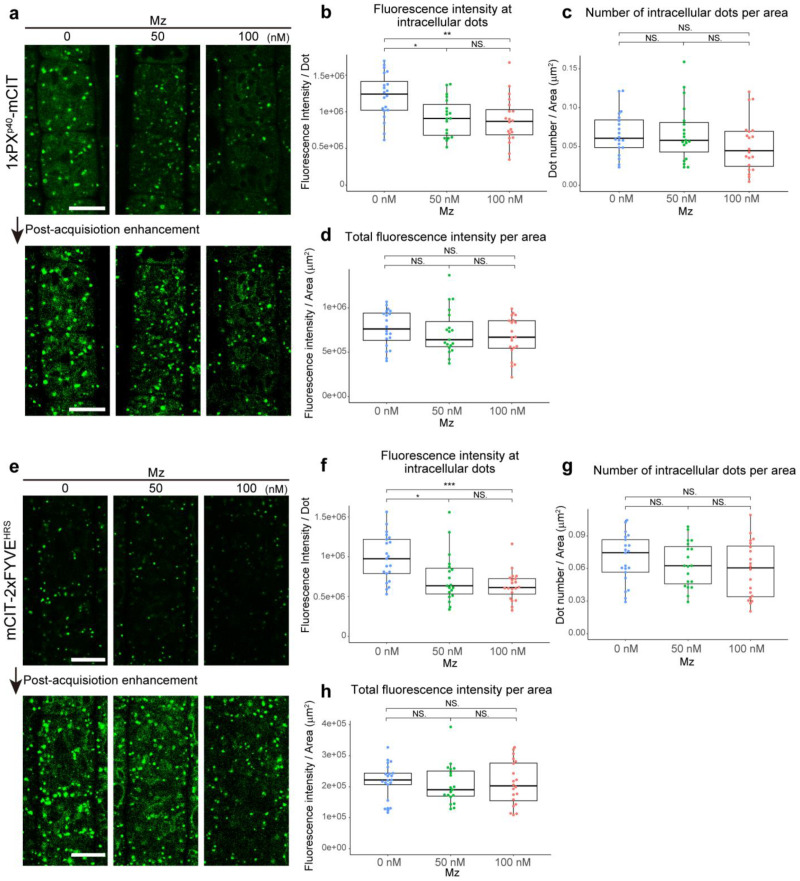
VLCFAs mediate intracellular PI3P homeostasis. (**a**,**e**) Confocal images of root epidermal cells displaying intracellular structures (green dots) labeled by either the PI3P biosensor 1xPX^P40^-mCIT (**a**) or the PI3P biosensor mCIT-2xFYVE^HRS^ (**e**) in control condition (0 nM), upon 50 nM metazachlor (Mz) or upon 100 nM Mz. For all the quantification, we strictly kept the same acquisition parameters between 0, 50 or 100 nM Mz treatments. The fluorescence signal at the intracellular dots decreased upon Mz treatment (upper panels), but the dotty structures were still able to be seen by post-acquisition enhancement (lower panels). (**b**,**f**) The fluorescence intensity of PI3P biosensors decreased at intracellular compartments upon 50 nM Mz or 100 nM Mz. (**c**,**g**) While tending towards a decrease, the number of intracellular dots labeled by PI3P was not significantly different between control condition and 50 nM Mz or 100 nM Mz treatment. (**d**,**h**) The general fluorescence intensity normalized by the area in µm^2^ was not different between control condition and 50 nM Mz or 100 nM Mz treatment. n = more than 19 roots for each condition. Statistics were done by two-sided Dwass–Steel–Critchlow–Flinger multiple comparison test with Monte Carlo method (10,000 iterations); * *p* < 0.05, ** *p* < 0.01, *** *p* < 0.005; NS: non-significant. Scale bars, 10 µm.

**Figure 2 ijms-22-08450-f002:**
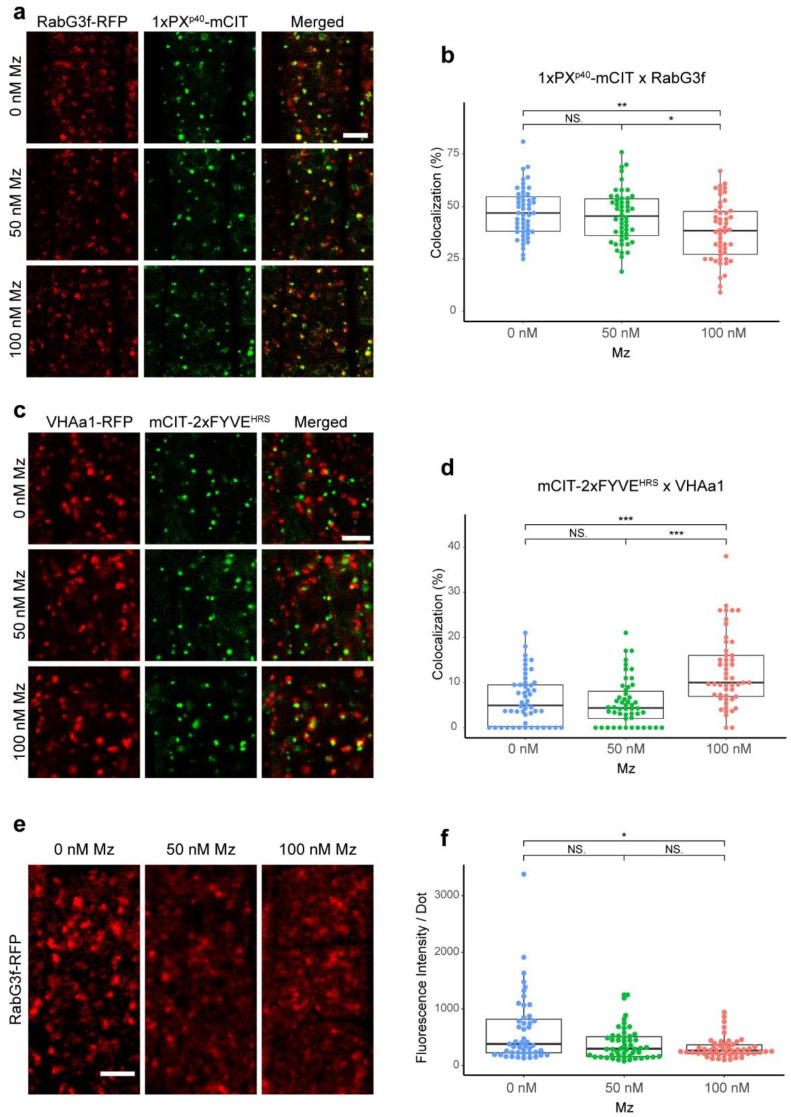
VLCFAs are involved in the relative abundance of PI3P between TGN and MVBs. (**a**) Confocal images of root epidermal cells expressing the MVB marker RAB-G3f-mRFP (red dots) and the PI3P 1xPX^P40^-mCIT-labeled structures (green dots) in control condition (0 nM), upon 50 nM metazachlor (Mz) or upon 100 nM Mz. (**b**) Co-localization analysis between PI3P-labeled vesicles and MVBs revealed that PI3P was less localized at MVBs upon 100 nM Mz. (**c**) Confocal images of root epidermal cells expressing the TGN marker VHA-a1-mRFP and the PI3P marker mCIT-2xFYVE^HRS^ in control condition (0 nM Mz), upon 50 nM Mz or upon 100 nM Mz. (**d**) Co-localization analysis between PI3P-labeled vesicles and TGN revealed that PI3P was more localized at TGN upon 100 nM Mz. (**e**) Confocal images of root epidermal cells expressing RAB-G3f-mRFP in control condition (0 nM), upon 50 nM Mz or upon 100 nM Mz. (**f**) In contrast to co-localization analyses, to quantify the fluorescence intensities at intracellular dots we strictly kept the same acquisition parameters between 0, 50 or 100 Mz treatments. We calculated a significant decrease of RAB-G3f signal intensity at MVBs upon 100 nM Mz. n = more than 47 areas distributed over 10 roots. Statistics were done by two-sided Dwass–Steel–Critchlow–Flinger multiple comparison test with Monte Carlo method (10,000 iterations); * *p* < 0.05, ** *p* < 0.01, *** *p* < 0.005; NS: non-significant. Scale bars, 5 µm.

**Figure 3 ijms-22-08450-f003:**
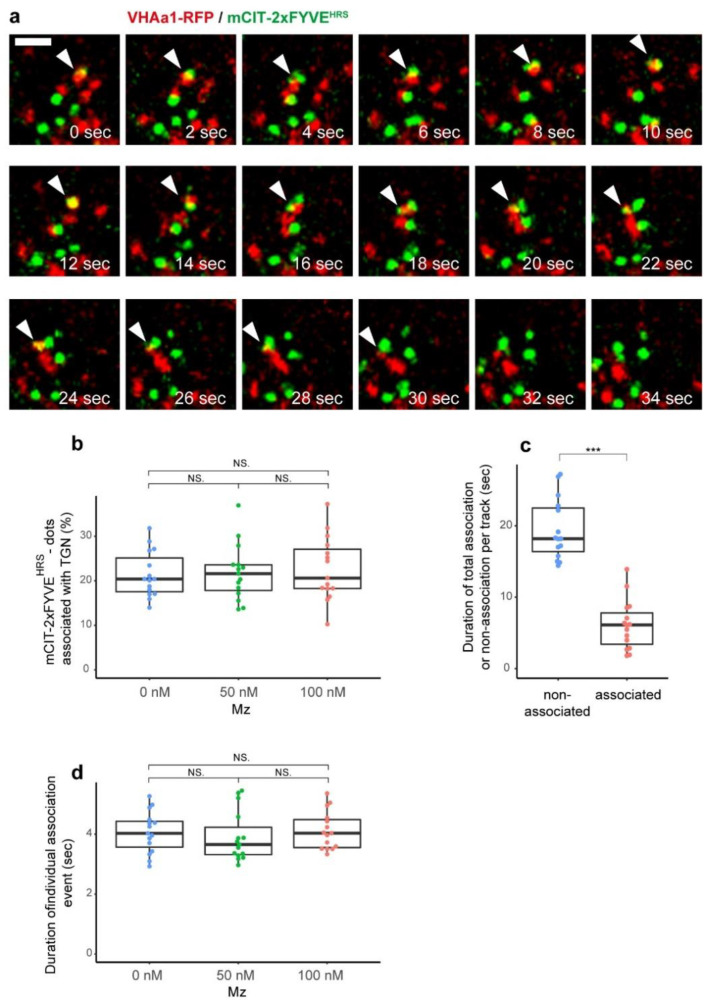
TGN and MVBs transiently associate together but VLCFAs are not involved in this process. (**a**) Representative example of a confocal Airyscan sequence of an association event in root epidermal cells expressing the TGN marker VHA-a1-mRFP (red dots) and the PI3P mCIT-2xFYVE^HRS^-labeled structures (green dots) in control condition. Associations (as between 2 and 10 s) as well as some co-localizations (as at 12 s) are observed. The white arrowheads indicate the association events. Sometimes, several PI3P-positive structures associate with one TGN, like at 22 s. Eventually, the compartments dissociate. (**b**) Quantification of the number of PI3P-positive structures that associate with TGN at one-time point; no significant differences were observed between control condition (0 nM), upon 50 nM metazachlor (Mz) or upon 100 nM Mz. (**c**) Quantification of the duration (in seconds) of the tracking of either non TGN-associated (average time of 19 s) or TGN-associated PI3P-positive structures (average time of 6 s). (**d**) The duration time of the TGN-associated PI3P-positive structure was not modified upon 50 nM Mz or upon 100 nM Mz. n = 15 roots for each condition. Statistics were done by two-sided Dwass–Steel-Critchlow-Flinger multiple comparison test with Monte Carlo method (10,000 iterations). *** *p* < 0.005; NS: non-significant. Scale bar, 2 µm.

**Figure 4 ijms-22-08450-f004:**
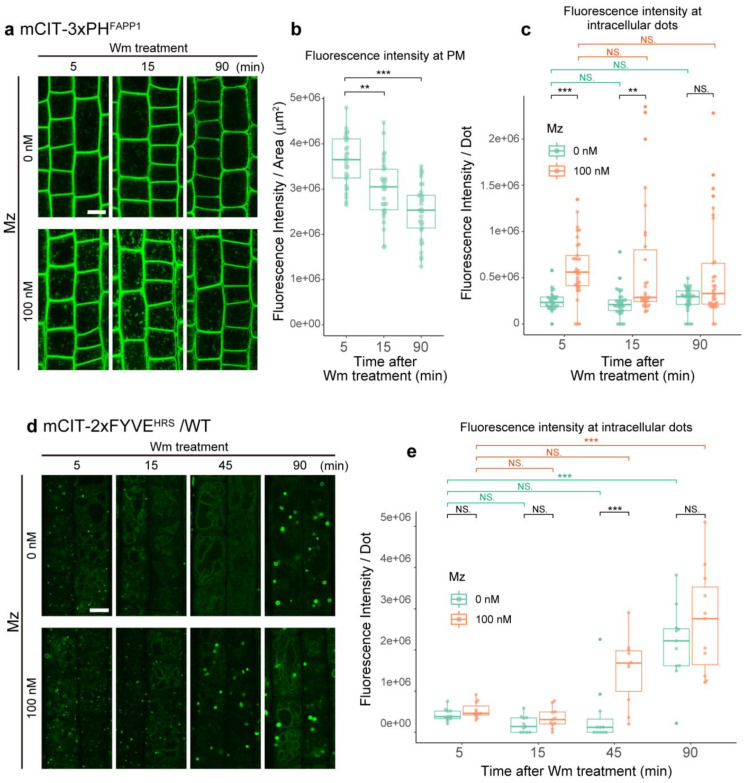
Effects of wortmannin treatment on PI4P and PI3P. (**a**) Root epidermal cells displaying PI4P membrane structures (TGN: green dots, and plasma membrane) labeled by the biosensor mCIT-3xPH^FAPP1^ during wortmannin (Wm) treatment (5, 15 and 90 min) of seedlings grown on 0 or 100 nM metazachlor (Mz)-containing medium. (**b**) Quantification of fluorescence intensity at the PM of the cells upon Wm treatment alone (n = more than 30 cells distributed over at least 10 roots for each condition). The PI4P pool at the PM was decreased significantly. (**c**) Quantification of the fluorescence intensity at intracellular dots (n = more than 30 cells distributed over at least 10 roots for each condition). The PI4P pool accumulated at intracellular dots by Mz was not significantly decreased upon Wm treatment. (**d**) Wild-type (WT) root epidermal cells expressing the PI3P mCIT-1xFYVEHRS biosensor during Wm treatment (5, 15 and 90 min) of seedlings grown on 0 or 100 nM Mz-containing medium. (**e**) Quantification of the fluorescence intensity at intracellular dots showing non-significant differences between 5, 15 and 45 min and high increase at 90 min of Wm treatment in control seedlings and high increase already after 45 min of Wm upon Mz treatment. Statistics were done by two-sided Dwass–Steel–Critchlow–Flinger multiple comparison test with Monte Carlo method (10,000 iterations) for comparing between different time points, and two-sided Wilcoxson’s rank-sum test for comparing between different Mz treatments. ** *p* < 0.01, *** *p* < 0.005; NS: non-significant. Scale bars, 10 µm.

**Figure 5 ijms-22-08450-f005:**
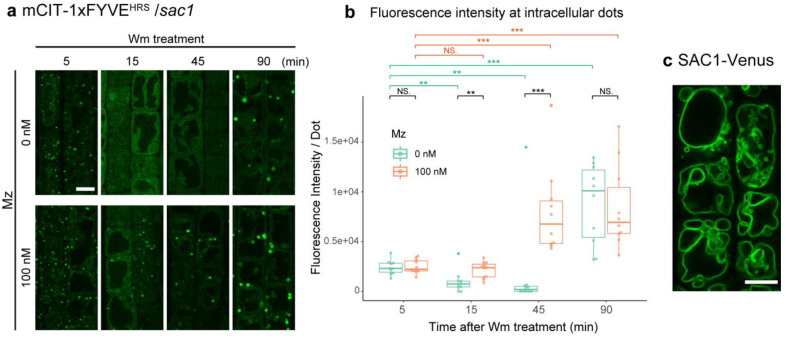
SAC1 is not involved in VLCFA-mediated PI3P homeostasis at TGN and MVBs. (**a**) *sac1* mutant root epidermal cells displaying PI3P intracellular structures (green dots)labeled by the mCIT-1xFYVE^HRS^ biosensor during Wm treatment (5, 15 and 90 min) of seedlings grown on 0 or 100 nM Mz-containing medium. (**b**) Quantification of the fluorescence intensity at intracellular dots showing significant differences between 5, 15 and 45 min and high increase at 90 min of Wm treatment in control seedlings and high increase already after 45 min of Wm upon Mz treatment. (**c**) Intracellular localization of SAC1-Venus displaying localization at vacuoles. n = more than 10 roots for each condition. Statistics were done by two-sided Dwass–Steel–Critchlow–Flinger multiple comparison test with Monte Carlo method (10,000 iterations) for comparing between different timepoints, and two-sided Wilcoxson’s rank-sum test for comparing between different Mz treatments. ** *p* < 0.01, *** *p* < 0.005; NS: non-significant. Scale bars, 10 µm.

**Figure 6 ijms-22-08450-f006:**
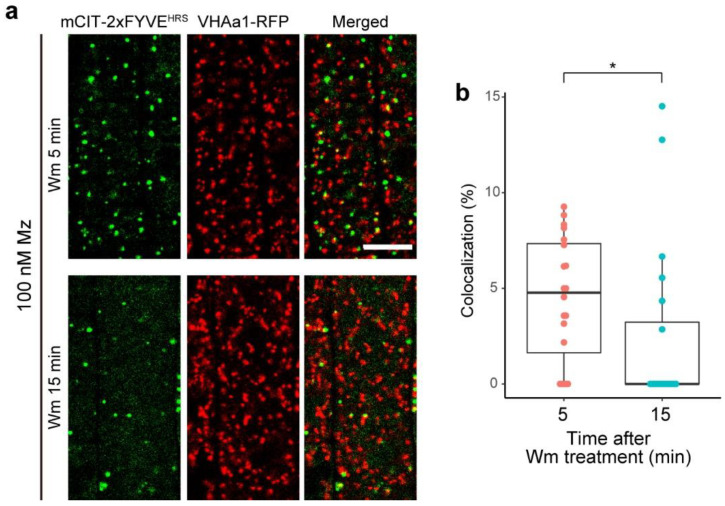
PI3P might be transferable from TGN to MVBs. (**a**) Root epidermal cells expressing the TGN marker VHA-a1-mRFP (red dots) and the PI3P mCIT-2xFYVE^HRS^-labeled structures (green dots) in wild-type seedlings grown on 100 nM metazachlor (Mz) during wortmannin (Wm) treatment (5 and 15 min). (**b**) Quantification of the co-localization level between PI3P biosensor and the TGN showing a decrease of PI3P at TGN between 5 and 15 min of Wm treatment in Mz-grown seedlings. n = 20 roots for each condition. Statistics were done by two-sided Wilcoxson’s rank-sum test. * *p* < 0.05. Scale bar, 10 µm.

## Data Availability

Data supporting the findings of this work are available within the paper. All other datasets and plant materials generated and analyzed during the current study are available from the corresponding author upon request. The new ImageJ plugin “Detect, Track and Co-localize (DTC)” is available at the following address: https://github.com/fabricecordelieres/IJ-Plugin_DTC, accessed on 27 July 2021.

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
