# Peer review of "Inhibition of Very Long Chain Fatty Acids Synthesis Mediates PI3P Homeostasis at Endosomal Compartments"

_ijms, 2021, doi:10.3390/ijms22168450_

Round 1
Reviewer 1 Report
In this manuscript, the authors analyzed effects of metazachlor, an inhibitor of VLCFA synthesis, on subcellular localization of PIP3 in plant cells. The authors showed that metazachlor treatment resulted in decrease of the PIP3 biosensor signals in the intracellular dots. Colocalization analyses using the MVB and TGN markers suggested involvement of VLCFA in distribution of PIP3 between TGN and MVB. The authors also showed localization of PIP3 in the MVBs after wortmannin treatment. This process was enhanced by metazachlor treatment. The presented results are potentially interesting. However, mechanisms underlying these observations are not clearly shown.
Major points
- Mechanisms which explain decrease of the PIP3 biosensor signals in the intracellular dots are not clear. We can see intracellular dots with high PIP3 biosensor signals in the metazachlor-treated cells. Have the authors observed increased PIP3 biosensor signals in the TGN-marker positive structure?
- The authors showed that metazachlor treatment enhanced localization of PIP3 in the presence of wortmannin. The results are not consistent with the above observation that colocalization of the PIP3 biosensor signals and the TGN marker increased by metazachlor treatment. The authors should explain these differences in more detail.
- The authors did not demonstrate that the PIP3 biosensor signal-positive structures observed in the wortmannin-treated cells are the MVBs in this manuscript. Colocalization experiments with the MVB marker as shown in Fig. 6 are necessary.
Author Response
Manuscript ID ijms-1293381 – cover letter with responses to reviewers’ comment
REV #1
In this manuscript, the authors analyzed effects of metazachlor, an inhibitor of VLCFA synthesis, on subcellular localization of PIP3 in plant cells. The authors showed that metazachlor treatment resulted in decrease of the PIP3 biosensor signals in the intracellular dots. Colocalization analyses using the MVB and TGN markers suggested involvement of VLCFA in distribution of PIP3 between TGN and MVB. The authors also showed localization of PIP3 in the MVBs after wortmannin treatment. This process was enhanced by metazachlor treatment. The presented results are potentially interesting. However, mechanisms underlying these observations are not clearly shown.
We would like to thank this reviewer for his appreciation of our work and his constructive comments. The main conceptual advance of this manuscript is the description of a novel aspect of lipid homeostasis between VLCFA and PI3P at the TGN-MVBs interface. The identification of the underlying mechanisms would undoubtedly be very interesting but would certainly requires extensive additional work and is thus beyond the scope of this manuscript.
Major points
- Mechanisms which explain decrease of the PIP3 biosensor signals in the intracellular dots are not clear. We can see intracellular dots with high PIP3 biosensor signals in the metazachlor-treated cells. Have the authors observed increased PIP3 biosensor signals in the TGN-marker positive structure?
We agree that discovering the mechanisms explaining the decrease of PI3P in intracellular compartments upon metazachlor treatment would be greatly interesting. However, this is beyond the scope of this study.
The co-localization method we are using is an object-based approach that accounts the morphology of the compartments, independently from the intensities at these compartments. This approach has the advantage to be more stringent than a classical Pearson coefficient which is based on intensities overlay. From our co-localization analyses, we could detect that more TGN compartments were PI3P-positive upon metazachlor treatment. However, we agree with this reviewer that getting more PI3P-positive TGN compartments does not necessarily means a higher quantity of PI3P in each TGN compartments in average. Unfortunately, as our co-localization method is object-based we set-up the acquisition parameters so that the intensity at endomembrane compartments remain similar between untreated and metazachlor-treated samples to avoid any intensity-related variation that would bias the morphology of the compartments. Thus, we cannot quantify the intensities of PI3P biosensors at TGN from our co-localization pictures. We thank the reviewer to raise this issue, this will help us in the future to design experiments where both object-based co-localization analyses and fluorescence intensity quantification could be done at the same time on the same batch of data.
However, to address the issue raised by this reviewer we modify the text to be more cautious in the interpretations of the co-localization and fluorescence intensities data. We also changed the title.
- The authors showed that metazachlor treatment enhanced localization of PIP3 in the presence of wortmannin. The results are not consistent with the above observation that colocalization of the PIP3 biosensor signals and the TGN marker increased by metazachlor treatment. The authors should explain these differences in more detail.
As rightly pointed out in the first comment of this reviewer, increased co-localization of PI3P biosensor at the TGN and increased average of fluorescence intensities of PI3P biosensor at the TGN are not necessarily correlated (due that the co-localization method is object-based), and thus not directly comparable. However, the results we got are not contradictory with our hypothesis that the TGN could serve as a small sink of PI3P. Previous work has shown that upon wortmannin, homotypic fusion of MVBs as well as heterotypic fusion of the TGN and MVBs occur [1]. Given that the PI3K complex is located at MVBs and that more TGNs contain PI3P biosensor upon metazachlor, it is possible that the increased number of TGN containing PI3P upon wortmannin+metazachlor accounts for the “resistance” of metazachlor-treated seedlings to PI3P-loss upon wortmannin treatment. Consistently with this hypothesis, the decrease of PI3P biosensor at the TGN upon wortmannin+metazachlor (Fig. 6) indicates that the TGN could indeed serve as a sink through heterotypic fusion between the TGN and MVBs to create donut-like structures upon wortmannin treatment. The wortmannin approach was a first indication on potential interaction between the TGN and MVBs. This is why we conducted Airyscan experiments to be able to indeed detect TGN-MVBs associations, which have never been described before. Although we could detect significant lasting TGN-MVBs associations, metazachlor did not interfere with those indicating that PI3P might be transferred to MVBs maybe more through maturation than membrane associations. We are sorry if our reasoning was not clear enough on the wortmannin treatment. Thus, we revised the discussion of this manuscript to be more cautious on these results.
- The authors did not demonstrate that the PIP3 biosensor signal-positive structures observed in the wortmannin-treated cells are the MVBs in this manuscript. Colocalization experiments with the MVB marker as shown in Fig. 6 are necessary.
Our point in Fig. 6 was not to visualize PI3P at MVBs upon wortmannin treatment, this was already described before [1,2]. Our point was to visualize the small pool of PI3P at the TGN. Originally we tried the wortmannin+metazachlor treatment on the 1xPX40-mCIT line crossed with RabG3f-RFP but unfortunately 1xPX40-mCIT became too cytosolic, as compared to mCIT-2xFYVEHRS, and did not remain on dotty structures, even with metazachlor treatment. We also tried with the mCIT-2xFYVEHRS line crossed with SNX1-RFP but the SNX1 signal was too weak to detect MVBs. Unfortunately, we were unsuccessful in getting the mCIT-2xFYVEHRS crossed line with RabG3f-RFP. In principle we agree with this reviewer that this would have been a nice control but PI3P has already been shown to locate at donut-like structures upon wortmannin and these structures have been extensively characterized as being a fused agglomeration of mostly MVBs and to a lesser extend post-Golgi compartments [1–3].
REFERENCES
- Takáč, T.; Pechan, T.; Šamajová, O.; Ovečka, M.; Richter, H.; Eck, C.; Niehaus, K.; Šamaj, J. Wortmannin treatment induces changes in arabidopsis root proteome and Post-Golgi compartments. J. Proteome Res. 2012, 11, 3127–3142, doi:10.1021/pr201111n.
- Simon, M.L.A.; Platre, M.P.; Marquès-Bueno, M.M.; Armengot, L.; Stanislas, T.; Bayle, V.; Caillaud, M.-C.; Jaillais, Y. A PtdIns(4)P-driven electrostatic field controls cell membrane identity and signalling in plants. Nat. Plants 2016, 2, 16089, doi:10.1038/nplants.2016.89.
- Wang, J.; Cai, Y.; Miao, Y.; Lam, S.K.; Jiang, L. Wortmannin induces homotypic fusion of plant prevacuolar compartments. J. Exp. Bot. 2009, 60, 3075–3083, doi:10.1093/jxb/erp136.
Reviewer 2 Report
This very nice manuscript by Ito et al. entitled “Inhibition of very long chain fatty acids synthesis mediate PI3P homeostasis between the trans-Golgi Network and multi-vesicular bodies.” focuses on how the VLCFAs of sphingolipids function in controlling the distribution of PI3P between the TGN and the MVBs. Using biosensors to detect PI3P and a chemical that reduces the chain length of the VLCFAs, the authors could show that VLCFAs have an impact on the relative distribution of PI3P between the TGN and MVBs. Although there are association events between PI3P and the TGN these are transient and not affected by VLCFAs. Next, using a pharmacological approach they depleted the pool of PI3P at MVBs by inhibiting PI3K and could show that metazachlor could act in a PI3P compensation mechanism, which is not dependent on PI(3,5)P2 degradation.
This manuscript is of great interest for plant scientists and overall, the manuscript is easy to read and the data clearly presented. At this point, however, there are some points that should be clarified as listed below.
Major points:
Figure 2: Although the data is very conclusive it is not clear why the authors used two different types of biosensor to detect PI3P on MVBs or on TGN. In my opinion, to be able assess the PI3P localization pattern correctly and to be able to compare the co-localization at the TGN to that of the MVBs, it would be more convincing if the authors used the same biosensor crossed once with the TGN marker and once with the MVB marker.
Figure 3: Once again, the data presented is very nice but to fully be able to say that (Lines 197-200) “TGN and PI3P-positive MVBs can remain associated for an average of time of 6 seconds (Fig. 3c). In comparison, MVBs that are not associated with the TGN could be followed unassociated for an average of time of 19 seconds (Fig. 3c).” the same experiments should be conducted with PI3P sensor and the MVB marker. Otherwise, this part of the paper has to be rephrased as the data presented in Figure 3 only allows does assessment of the association of PI3P with the TGN but nor with MVBs.
Minor points:
Figure 1: For the quantification of the global intensity of the fluorescence of the biosensors, both cytoplasmic and intracellular structures were quantified and there was no significant difference observed. Nevertheless, the fluorescence in the intracellular structures was reduced (Fig. 1a, b, e, f), does this imply that there is more cytoplasmic signal and what would this suggest for the PI3P sensors?
Lines 314-315: “Given that we observed an increase of PI3P biosensors at the TGN upon metazachlor, one possibility is that the pool of PI3P at the TGN serves as a sink that recovers PI3P upon wortmannin.” Please elaborate a bit further on this assumption, either here or in the discussion.
Some typos:
Lines 83-84: Similarly, In Arabidopsis, fluorescent biosensors has revealed that PI3P
Line 297-8 SAC1 was found to localize at the TGN and, where it could regulate PI3P homeostasis at the TGN/MVBs interface [41].
Author Response
Manuscript ID ijms-1293381 – cover letter with responses to reviewers’ comment
REV #2
This very nice manuscript by Ito et al. entitled “Inhibition of very long chain fatty acids synthesis mediate PI3P homeostasis between the trans-Golgi Network and multi-vesicular bodies.” focuses on how the VLCFAs of sphingolipids function in controlling the distribution of PI3P between the TGN and the MVBs. Using biosensors to detect PI3P and a chemical that reduces the chain length of the VLCFAs, the authors could show that VLCFAs have an impact on the relative distribution of PI3P between the TGN and MVBs. Although there are association events between PI3P and the TGN these are transient and not affected by VLCFAs. Next, using a pharmacological approach they depleted the pool of PI3P at MVBs by inhibiting PI3K and could show that metazachlor could act in a PI3P compensation mechanism, which is not dependent on PI(3,5)P2 degradation.
This manuscript is of great interest for plant scientists and overall, the manuscript is easy to read and the data clearly presented. At this point, however, there are some points that should be clarified as listed below.
We would like to thank this reviewer for his appreciation of our work and his constructive comments.
Major points:
Figure 2: Although the data is very conclusive it is not clear why the authors used two different types of biosensor to detect PI3P on MVBs or on TGN. In my opinion, to be able assess the PI3P localization pattern correctly and to be able to compare the co-localization at the TGN to that of the MVBs, it would be more convincing if the authors used the same biosensor crossed once with the TGN marker and once with the MVB marker.
We completely agree with this reviewer. The best would have been to get both PI3P biosensor lines, mCIT-2xFYVEHRS or 1xPX40-mCIT, crossed with either VHAa1-RFP or RabG3f-RFP. Previously published co-localization analyses have shown that both biosensors mainly locate at MVBs and sparsely at the TGN [1]. Originally we preferred to used mCIT-2xFYVEHRS given that this PI3P biosensor remains focus on dotty intracellular structures while 1xPX40-mCIT additionally labels the vacuoles with the consequence to dilute more this biosensor over the intracellular membranes. Unfortunately, we were unsuccessful in getting the mCIT-2xFYVEHRS crossed line with RabG3f-RFP. We also tried with the mCIT-2xFYVEHRS line crossed with SNX1-RFP but the SNX1 signal was too weak to detect MVBs. However, we completely agree that in this case the manuscript can be revised to avoid the direct comparison of PI3P localization pattern at the TGN and MVBs. We revised the text accordingly as our interest and the main message of this manuscript were more on the TGN than MVBs anyway. We also changed the title.
Figure 3: Once again, the data presented is very nice but to fully be able to say that (Lines 197-200) “TGN and PI3P-positive MVBs can remain associated for an average of time of 6 seconds (Fig. 3c). In comparison, MVBs that are not associated with the TGN could be followed unassociated for an average of time of 19 seconds (Fig. 3c).” the same experiments should be conducted with PI3P sensor and the MVB marker. Otherwise, this part of the paper has to be rephrased as the data presented in Figure 3 only allows does assessment of the association of PI3P with the TGN but nor with MVBs.
We agree with this reviewer but we would like to point that PI3P mainly localizes at MVBs, this has been shown in several studies [1–4]. However, we revised our text as asked by the reviewer to be more careful on what has been demonstrated before and what our results show in this study.
Minor points:
Figure 1: For the quantification of the global intensity of the fluorescence of the biosensors, both cytoplasmic and intracellular structures were quantified and there was no significant difference observed. Nevertheless, the fluorescence in the intracellular structures was reduced (Fig. 1a, b, e, f), does this imply that there is more cytoplasmic signal and what would this suggest for the PI3P sensors?
Yes, the PI3P biosensors signal gets more cytosolic upon metazachlor although this is difficult to see by eyes in Fig. 1 since the signal is diffused into the cytosol. Consistently, our results show no differences when we quantified the overall biosensor signals in a given squared area (cytosol+intracellular structures) (Fig. 1d, h). These results also indicate that there are no differences in the expression level of the biosensors between control and metazachlor-treated seedlings. We quantified a decrease of PI3P biosensor at intracellular structures which actually corresponds to a disassociation of the biosensors from these structures. However, the number of these structures labeled by the PI3P biosensors was not altered, showing that the effect we see is directly related to the amount of PI3P biosensors that binds intracellular structures, and not to less structures labeled by PI3P.
Lines 314-315: “Given that we observed an increase of PI3P biosensors at the TGN upon metazachlor, one possibility is that the pool of PI3P at the TGN serves as a sink that recovers PI3P upon wortmannin.” Please elaborate a bit further on this assumption, either here or in the discussion.
Yes, this has also been pointed out by reviewer #1.
Previous work has shown that upon wortmannin, homotypic fusion of MVBs as well as heterotypic fusion of the TGN and MVBs occur [5]. Given that the PI3K complex is located at MVBs and that more TGNs contain PI3P biosensor upon metazachlor, it is possible that the increased number of TGN containing PI3P upon wortmannin+metazachlor accounts for the “resistance” of metazachlor-treated seedlings to PI3P-loss upon wortmannin treatment. Consistently with this hypothesis, the decrease of PI3P biosensor at the TGN upon wortmannin+metazachlor (Fig. 6) indicates that the TGN could indeed serve as a sink through heterotypic fusion between the TGN and MVBs to create donut-like structures upon wortmannin treatment. The wortmannin approach was a first indication on potential interaction between the TGN and MVBs. This is why we conducted Airyscan experiments to be able to indeed detect TGN-MVBs associations, which have never been described before. Although we could detect significant lasting TGN-MVBs associations, metazachlor did not interfere with those indicating that PI3P might be transferred to MVBs maybe more through maturation than membrane associations. We are sorry if our reasoning was not clear enough on the wortmannin treatment. Thus, we revised the discussion of this manuscript to be more cautious on these results.
Some typos:
Lines 83-84: Similarly, In Arabidopsis, fluorescent biosensors has revealed that PI3P
This has been corrected
Line 297-8 SAC1 was found to localize at the TGN and, where it could regulate PI3P homeostasis at the TGN/MVBs interface [41].
This has been corrected
REFERENCES
- Simon, M.L.A.; Platre, M.P.; Assil, S.; van Wijk, R.; Chen, W.Y.; Chory, J.; Dreux, M.; Munnik, T.; Jaillais, Y. A multi-colour/multi-affinity marker set to visualize phosphoinositide dynamics in Arabidopsis. Plant J. 2014, 77, 322–337, doi:10.1111/tpj.12358.
- Vermeer, J.E.M.; Van Leeuwen, W.; Tobeña-Santamaria, R.; Laxalt, A.M.; Jones, D.R.; Divecha, N.; Gadella, T.W.J.; Munnik, T. Visualization of PtdIns3P dynamics in living plant cells. Plant J. 2006, 47, 687–700, doi:10.1111/j.1365-313X.2006.02830.x.
- Gao, C.; Luo, M.; Zhao, Q.; Yang, R.; Cui, Y.; Zeng, Y.; Xia, J.; Jiang, L. A Unique plant ESCRT component, FREE1, regulates multivesicular body protein sorting and plant growth. Curr. Biol. 2014, 24, 2556–2563, doi:10.1016/j.cub.2014.09.014.
- Singh, M.K.; Krüger, F.; Beckmann, H.; Brumm, S.; Vermeer, J.E.M.; Munnik, T.; Mayer, U.; Stierhof, Y.-D.; Grefen, C.; Schumacher, K.; et al. Protein Delivery to Vacuole Requires SAND Protein-Dependent Rab GTPase Conversion for MVB-Vacuole Fusion. Curr. Biol. 2014, 24, 1383–1389, doi:10.1016/j.cub.2014.05.005.
- Takáč, T.; Pechan, T.; Šamajová, O.; Ovečka, M.; Richter, H.; Eck, C.; Niehaus, K.; Šamaj, J. Wortmannin treatment induces changes in arabidopsis root proteome and Post-Golgi compartments. J. Proteome Res. 2012, 11, 3127–3142, doi:10.1021/pr201111n.
Reviewer 3 Report
Dear Authors,
The article “Inhibition of very long chain fatty acids synthesis mediates PI3P homeostasis between the trans-Golgi Network and multivesicular bodies" by Y. Ito et al., aimed to characterize the role of lipid biosynthetic processes (particular contribution of sphingolipids with very long fatty acid chains) on major pathways regulating Phosphatidylinositol-3-phosphate (PI3P) levels at early endosome/trans-Golgi Network (TGN) compartments in plant cells. In an attempt to unveil the mechanism that regulates s distribution of PI3P between TGN/MVBs membranes authors utilized Arabidopsis (plant) model. Experiments and assays used in the current study are up-to-date and well established in the membrane trafficking field and lipidomics.
Here is the list of my main concerns and comments about the current paper:
Question 1. In Figure 1 the authors demonstrate that treatment with metazachlor causes significant decrease in intensity from PI3P biosensors (1xPXP40-mCIT and mCIT-2xFYVEHRS) while the distribution of biosensors within intracellular compartments are not affected. The difference is obvious, nevertheless it would be nice to show that expression levels of biosensors in cells between control and inhibitor are comparable (for example Figure 1A and E difference is obvious, while biosensor (green channel) columns Figures 2A and C do not show visible differences).
Question 2. I'm wondering what could explain the difference in effective concentrations in Figure 1a,b and 2a,b,c,d. To be more clear: Intensity of PI3P biosensors significantly drops at 50nM inhibitors while colocalization with VHAa1/lost colocalization with RabG3f appears at 100nM.
Question 3. Are effects of metazachlor on PI3P distribution mediated by changes in plant sterol levels in studies compartments? It would be nice to see data excluding effects on sterol levels in studies compartments.
Minor comments:
Scale bar is barely visible on Figure 4a,d.
Author Response
Manuscript ID ijms-1293381 – cover letter with responses to reviewers’ comment
REV #3
The article “Inhibition of very long chain fatty acids synthesis mediates PI3P homeostasis between the trans-Golgi Network and multivesicular bodies" by Y. Ito et al., aimed to characterize the role of lipid biosynthetic processes (particular contribution of sphingolipids with very long fatty acid chains) on major pathways regulating Phosphatidylinositol-3-phosphate (PI3P) levels at early endosome/trans-Golgi Network (TGN) compartments in plant cells. In an attempt to unveil the mechanism that regulates s distribution of PI3P between TGN/MVBs membranes authors utilized Arabidopsis (plant) model. Experiments and assays used in the current study are up-to-date and well established in the membrane trafficking field and lipidomics.
We would like to thank this reviewer for his appreciation of our work and his constructive comments.
Here is the list of my main concerns and comments about the current paper:
Question 1. In Figure 1 the authors demonstrate that treatment with metazachlor causes significant decrease in intensity from PI3P biosensors (1xPXP40-mCIT and mCIT-2xFYVEHRS) while the distribution of biosensors within intracellular compartments are not affected. The difference is obvious, nevertheless it would be nice to show that expression levels of biosensors in cells between control and inhibitor are comparable (for example Figure 1A and E difference is obvious, while biosensor (green channel) columns Figures 2A and C do not show visible differences).
Yes, we agree with this reviewer. Actually, the PI3P biosensors signal gets more cytosolic upon metazachlor although this is difficult to see by eyes in Fig. 1 since the signal is diffused into the cytosol. Consistently, our results show no differences when we quantified the overall biosensor signals in a given squared area (cytosol+intracellular structures) (Fig. 1d, h). These results also indicate that there are no differences in the expression level of the biosensors between control and metazachlor-treated seedlings. We quantified a decrease of PI3P biosensor at intracellular structures which actually corresponds to a disassociation of the biosensors from these structures. However, the number of these structures labeled by the PI3P biosensors was not altered, showing that the effect we see is directly related to the amount of PI3P biosensors that binds intracellular structures, and not to less structures labeled by PI3P.
As for the comparison between Fig. 1 where the PI3P biosensors signal is dropping and Fig. 2 where the same signal is not dropping under the same conditions, we should precise here that these are two different types of experiments. In Fig. 1 we kept the acquisition parameters constant to quantify the fluorescence intensity at compartments. In Fig. 2, as our co-localization method is object-based we set-up the acquisition parameters so that the intensity at endomembrane compartments remains similar between untreated and metazachlor-treated samples to avoid any intensity-related variation that would bias the morphology of the compartments. Thus, the images in Fig. 2 do not reflect the effects of metazachlor treatment on the signal intensity of PI3P biosensors as they were designed for co-localization analyses.
Question 2. I'm wondering what could explain the difference in effective concentrations in Figure 1a,b and 2a,b,c,d. To be more clear: Intensity of PI3P biosensors significantly drops at 50nM inhibitors while colocalization with VHAa1/lost colocalization with RabG3f appears at 100nM.
The co-localization method we are using is an object-based approach that accounts the morphology of the compartments, independently from the intensities at these compartments. This approach has the advantage to be more stringent than a classical Pearson coefficient which is based on intensities overlay. Thus, from our co-localization analyses, we could detect that more TGN compartments were PI3P-positive upon metazachlor treatment. However, increased co-localization of PI3P biosensor at the TGN and increased average of fluorescence intensities of PI3P biosensor at the TGN are not necessarily correlated and thus not directly comparable. Still, the differences observed between the intensities at 50 nM and the co-localization at 100 nM could suggest the existence of two different processes acting on PI3P homeostasis. One could be the production and consumption of PI3P (the intensities as a readout), another one could be the localization balance between the TGN and MVBs (the co-localization as a readout).
We revised the text to be more cautious in the interpretations of the co-localization and fluorescence intensities data.
Question 3. Are effects of metazachlor on PI3P distribution mediated by changes in plant sterol levels in studies compartments? It would be nice to see data excluding effects on sterol levels in studies compartments.
The sterol composition of the total pool of membranes (microsomes) or immuno-purified TGN is not modified upon metazachlor, while the fatty acid profile is clearly altered (decrease in C24-fatty acids) in these membrane fractions [1,2]. We do not have these data for MVBs but sterols mainly localize to plasma membrane and TGN in Arabidopsis [3].
We introduced this in the discussion of the manuscript.
Minor comments:
Scale bar is barely visible on Figure 4a,d.
We changed the scale bar.
REFERENCES
- Wattelet-Boyer, V.; Brocard, L.; Jonsson, K.; Esnay, N.; Joubès, J.; Domergue, F.; Mongrand, S.; Raikhel, N.; Bhalerao, R.P.; Moreau, P.; et al. Enrichment of hydroxylated C24-and C26-acyl-chain sphingolipids mediates PIN2 apical sorting at trans-Golgi network subdomains. Nat. Commun. 2016, 7, doi:10.1038/ncomms12788.
- Ito, Y.; Esnay, N.; Platre, M.P.; Noack, L.; Menzel, W.; Claverol, S.; Moreau, P.; Jaillais, Y.; Boutté, Y. Sphingolipids mediate polar sorting of PIN2 through phosphoinositide consumption at the trans -Golgi Network. Nat. Commun. 2020, 12, 4267, doi:10.1101/2020.05.12.090399.
- Boutté, Y.; Frescatada-Rosa, M.; Men, S.; Chow, C.-M.; Ebine, K.; Gustavsson, A.; Johansson, L.; Ueda, T.; Moore, I.; Jürgens, G.; et al. Endocytosis restricts Arabidopsis KNOLLE syntaxin to the cell division plane during late cytokinesis. EMBO J. 2010, 29, doi:10.1038/emboj.2009.363.
Round 2
Reviewer 1 Report
The revised manuscript has been improved. The authors have adequately addressed to my earlier comments.
Reviewer 2 Report
Thank you. I am satisfied with the comments from the authors and recommend publication.